# Vegetation and vertebrate abundance as drivers of bioturbation patterns along a climate gradient

Diana Kraus[1]*, Roland Brandl[2], Sebastian Achilles[3], Jörg Bendix[3],
Paulina Grigusova[3], Annegret Larsen[4], Patricio Pliscoff[5], Kirstin Übernickel[6],
Nina Farwig[1]

1 Department of Biology, Conservation Ecology, University of Marburg, Marburg, Germany, 2 Department of Biology, Animal Ecology, University of Marburg, Marburg, Germany, 3 Department of Geography, Laboratory for Climatology and Remote Sensing, University of Marburg, Marburg, Germany, 4 Department of Environmental Sciences, Soil Geography and Landscape, Wageningen University & Research, Wageningen, Netherlands, 5 Department of Ecology and Biodiversity and Institute of Geography, Catholic University of Chile, Santiago, Chile, 6 Department of Geosciences, Earth System Dynamics, University of Tübingen, Tübingen, Germany

* diana.kraus@biologie.uni-marburg.de

**Data Availability Statement:** All relevant data are within the manuscript and its Supporting Information files.

## Abstract

Bioturbators shape their environment with considerable consequences for ecosystem processes. However, both the composition and the impact of bioturbator communities may change along climatic gradients. For burrowing animals, their abundance and composition depend on climatic and other abiotic components, with ants and mammals dominating in arid and semiarid areas, and earthworms in humid areas. Moreover, the activity of burrowing animals is often positively associated with vegetation cover (biotic component). These observations highlight the need to understand the relative contributions of abiotic and biotic components in bioturbation in order to predict soil-shaping processes along broad climatic gradients. In this study, we estimated the activity of animal bioturbation by counting the density of holes and the quantity of bioturbation based on the volume of soil excavated by bioturbators along a gradient ranging from arid to humid in Chile. We distinguished between invertebrates and vertebrates. Overall, hole density (no/ 100 m²) decreased from arid (raw mean and standard deviation for invertebrates: 14 ± 7.8, vertebrates: 2.8 ± 2.9) to humid (invertebrates: 2.8 ± 3.1, vertebrates: 2.2 ± 2.1) environments. However, excavated soil volume did not follow the same clear geographic trend and was 300-fold larger for vertebrates than for invertebrates. The relationship between bioturbating invertebrates and vegetation cover was consistently negative whereas for vertebrates both, positive and negative relationships were determined along the gradient. Our study demonstrates complex relationships between climate, vegetation and the contribution of bioturbating invertebrates and vertebrates, which will be reflected in their impact on ecosystem functions.

**Funding:** This study was funded by the German Science Foundation DFG Priority Program SPP 1803: EarthShape: Earth Surface Shaping by Biota, sub-project "Effects of bioturbation on rates of vertical and horizontal sediment and nutrient fluxes" [grant numbers BE1780/52-1, LA3521/1-1, FA 925/12-1, BR 1293-18-1].

**Competing interests:** The authors have declared that no competing interests exist.

# Introduction

Bioturbation, the biological reworking of soils and sediments [1, 2], shapes the environment and thus has considerable consequences for ecosystem processes [3] such as sediment transport, soil formation [4, 5], soil water cycles [6], litter decomposition [7], and nutrient availability [8, 9]. Soil excavating animals range from small invertebrates such as ants [10, 11] and earthworms [12, 13] to medium-sized vertebrates such as gophers [14, 15] and beavers [16, 17]. Generally, bioturbating animals have distinct adaptations to environmental conditions but recent studies reveal that bioturbating animals are intentionally able to modify their environment [18–20]. Thus, assessments of the relative contributions of bioturbators to soil-shaping processes across larger climate gradients, must consider both the composition of bioturbator communities and their relationships to the abiotic and biotic environment. Previous studies indicate that:

1. The abundance and composition of burrowing animal communities depend on climatic (= abiotic) factors such as temperature and humidity [21–26]. Ants and mammals are the most important bioturbators in semiarid and arid areas, and earthworms (Lumbricidae) dominate in humid areas [27]. Local soil characteristics affect bioturbating activity, which is highest after rainfall because the soil softens and the energy cost of digging is accordingly reduced [28] as shown for burrowing mammals [29] and in the nest site selection of ants [23].

2. Burrowing animals are closely associated with biotic components of the environment, especially vegetation which affects the abundance of bioturbators directly by providing food [30, 31] and indirectly by providing habitat [32]. In humid regions with dense vegetation cover, food resources are generally abundant and thus mammals have less need to dig for food. Vegetation also provides shelter further reducing the need to dig. In resource-limited environments, such as semi-arid and arid regions, the activity and quantity of bioturbating mammals correlate positively with vegetation cover, because of those animals' need to seek subterranean food and shelter [33]. By contrast, invertebrates such as earthworms do not rely on surface resources offered by vegetation cover as they live entirely belowground, where they feed on dead roots in the soil [34].

Those studies demonstrate, that both, abiotic and biotic components influence bioturbation patterns, with the relationships between bioturbators and their environment varying between animal groups [35]. Detailed insights into the relative contributions of those groups can be obtained by associating them with their burrows, such as based on the diameter of the holes they create. A previous study has collected data on burrowing animals along a climate gradient and used a threshold of 2.5 cm to differentiate between vertebrates and invertebrates [36].

However, most studies have thus far focused either on the burrowing activity and quantity of single species (mostly vertebrates), or on individual climatic regions [36]. Studies on the overall patterns of bioturbation along broad climatic gradients are rare. To close this research gap, we examined the interaction of abiotic and biotic components along a broad climatic and vegetational gradient in Chile. For this purpose, we measured the abundance of burrow entrances (hole density) and the amount of soil excavated by burrowing animals (excavated soil volume) as parameters for bioturbation activity and quantity across seasons. Taking into account the available literature, we hypothesized that:

H1: Bioturbating activity decreases from arid to humid regions because climate drives the abundance of burrowing animals and the contribution of invertebrates and vertebrates to bioturbation patterns.

H2: Seasonal changes affect bioturbation with a higher activity of burrowing animals during rainy seasons, when the soil is softer, and the energetic cost of digging is therefore reduced.

H3: With increasing vegetation cover, the bioturbating activity of many invertebrates (including most earthworms) decreases, due to the subterranean food supply provided by fine roots in the soil independent of soil surface vegetation, while that of vertebrates increases, due to the increased availability of food and shelter.

## Methods

### Study area

Our study was conducted at four sites representing a climate gradient along the coastal range of Chile (26˚S-38˚S), extending from an arid desert with a mean annual temperature of 16.8˚C and mean annual precipitation of 12 mm to a temperate humid rainforest with a mean annual temperature of 6.6˚C and mean annual precipitation of 1469 mm [37]: arid Atacama Desert, located in Pan de Azúcar National Park, semi-arid shrubland in the private reserve Santa Gracia, a Mediterranean forest in La Campana National Park and a humid rainforest in Nahuelbuta National Park. All approvals from the relevant authorities, i.e. the Chilean National Forest Commission (CONAF), were obtained in advance to our study and granted access to the research sites. In 2019, the year of our field campaigns, the mean temperature in the arid desert was 14.6˚C and the mean precipitation was 9.4 mm while in the humid rainforest, the mean temperature was 7.3˚C and the mean precipitation was 1885 mm [38].

To sample each research site representatively, we established 12 10 m × 10 m plots with a distance of at least 30 m between them during the first field campaign, conducted in autumn of the southern hemisphere (March to April 2019). In a second field campaign conducted in spring of the southern hemisphere (September to November 2019) we established eight additional plots at each site to cover possible variation, resulting in a total of 20 plots per site. The 20 plots per research site were evenly distributed across two opposing hillsides, 10 on the north- and 10 on the south-facing hillslope.

### Assessment of bioturbation activity and quantity

To evaluate bioturbation activity, we counted the number of all visually detectable burrow entrances on the soil surface (hole density) of each plot. We calculated the amount of soil excavated by burrowing animals (excavated soil volume) as an indicator of bioturbation quantity by using a caliper to measure the vertical ($d_v$) and horizontal ($d_h$) diameters. In addition, we defined the depth of each hole entrance ($d_e$) as the distance to the first barrier encountered by the caliper and measured this parameter. Raw data of burrow measurements can be obtained from S7 Table in S4 Appendix. Following [36, 39, 40], we calculated the (minimal) excavated soil volume assuming that the measured burrows were cone-shaped:

$$excavated\ soil\ volume = \frac{1}{3} * \left[\frac{d_v + d_h}{4}\right]^2 * \pi * d_e.$$

To distinguish between the burrows of invertebrates and vertebrates, burrows with a hole-entrance diameter < 2.5 cm were assumed to be created by invertebrates and burrows with a hole-entrance diameter ≥ 2.5 cm by vertebrates [36].

### Assessment of vegetation data

Vegetation cover was estimated using unmanned aerial vehicle (UAV) red green blue (RGB) images and land cover classification [41]. For each plot, we calculated the ratio of pixels

classified as any plant type (herbs, shrubs, cacti, trees) to the amount of all pixels. Following [42], the average elevation (hillside elevation) and the hillslope of each plot were estimated based on high resolution Lidar data [43].

## Statistical analyses

For the burrows of invertebrates and vertebrates we analyzed the allometric relationship between their depth ($d_e$) and their diameter (mean of $d_v$ and $d_h$). We regressed the mean diameter of the entrance versus the depth using the $log_{10}$-transformed values of both variables and then determining the slope. In an isometric relationship, the log-transformed variables should be linearly related to a slope of one [44]. Since diameter and depth were measured with roughly equal error, in addition to an ordinary least squares (OLS) regression, we estimated the slope using a reduced major axis (RMA) regression [45]. To assess a deviation from a slope of one, we used the offset argument available in most regression functions. With the diameter serving as the independent variable and the offset, the estimate tests for deviations from one. For the slope of the RMA regression, we used the standard error and a t-test to test for deviations from one. The same approach was applied to the regression between excavated soil volume and hole density.

To analyze the interaction of abiotic and biotic components in bioturbation activity and quantity, we applied generalized linear mixed effect models (GLMMs). We used hole density or excavated soil volume as response variables, site, season, hillside elevation and hillslope as abiotic fixed predictors and vegetation cover and animal group as biotic fixed predictors. The study plots were used as a random factor (Table 1). All data of the GLMM parameters can be obtained from S7 Table in S4 Appendix. We also included interaction terms between site and all other fixed predictor variables and between vegetation cover and taxon. We standardized the fixed predictor hillside elevation for each site because it varied and could not be assigned separately to each of the sites. We performed GLMMs for the 12 plots within each site (total of 48 plots) in the first field campaign, conducted in the southern-hemispheric autumn, and in the 20 plots within each site (total of 80 plots) during the second campaign conducted in the southern-hemispheric spring. Separation of the hole density of invertebrates and vertebrates resulted in 256 measurements ($2 \times (48 + 80)$). For the GLMM of the excavated soil volume, we $log_{10}$-transformed data for hole density and excavated soil volume to achieve normality of the residuals. For the $log_{10}$-transformation, we only considered plots with a hole density $> 0$ [no/ 100 m$^2$]. Thus, 46 plots without holes were not included in the GLMM for excavated soil volume, resulting in 210 valid measurements. Additionally, we integrated the interaction between hole density and taxon as another fixed predictor.

All statistical analyses were performed using the R statistical environment (version 1.3.1093). We used the *lmodel2* package [46] for OLS and RMA regression analysis. For the GLMM, we employed the *buildmer* function [47] of the *lme4* package [48] to perform backward stepwise selection. To determine the proportion of variation explained by the model in

**Table 1. Summary of all variables used in the GLMM.**

| Response variable | Abiotic fixed predictors | Biotic fixed predictors | Random factor |
|---|---|---|---|
| **Hole density or excavated soil volume** | Site | Vegetation cover | Plot number |
| | Season | | |
| | Hillside elevation | Animal group | |
| | Hillslope | | |

Depicted are the response variables, fixed predictors (abiotic and biotic) and the random factor.

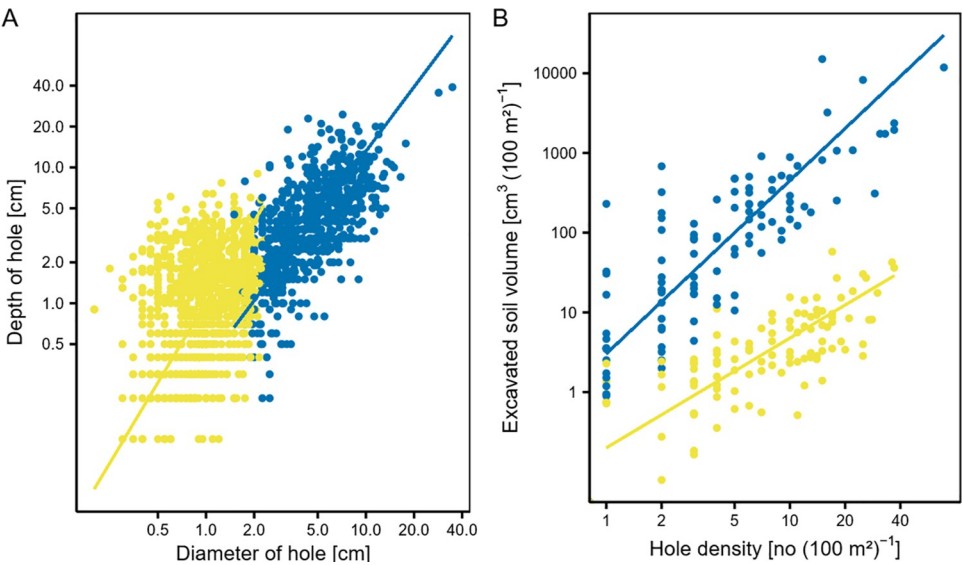

**Fig 1. Relationships between burrow parameters and bioturbation parameters created by burrowing invertebrates (yellow) and vertebrates (blue).** (A) Relationship between the depth and mean diameter of the holes, (B) relationship between the excavated soil volume and hole density. The regression lines are derived from the reduced major axis analysis. Note that both axes in (A) and (B) were $\log_{10}$-scaled. Data from both field campaigns and all sites were used.

total including fixed and random effects, we calculated R-squared for the fitted models using the *rsq* command from the *rsq* package [49]. We additionally performed an ANOVA between all possible combinations of fixed predictors retained within the fitted model to evaluate the significance of certain combinations between predictors using the *anova* command and performing a Chi-square test [50].

## Results

First, we investigated the allometric relationships by examining the relationship between the mean diameter and mean depth of the burrows and between the hole density and excavated soil volume. While the respective estimates of the slope are presented herein, our focus is on the RMA slopes. For both, invertebrates and vertebrates, the slopes showed a positive allometric relationship (Fig 1A, Table 2) that was maintained also in the single-season analysis (S1 Fig, S2 Table in S2 Appendix). However, note that the statistical tests evaluating burrow characteristics and the excavated soil volume were not strictly independent, as the former parameter was used to calculate the latter.

Hole density was always greater for invertebrates than for vertebrates (Fig 2A). For invertebrates, hole density decreased continuously from the arid site Pan de Azúcar (raw mean and standard deviation: $14 \pm 7.8$ no/ 100 m$^{-2}$) to the humid site Nahuelbuta ($2.8 \pm 3.1$ no/ 100 m$^{-2}$) while hole density for vertebrates was highest in the semi-arid site Santa Gracia ($9.1 \pm 9.7$ no/ 100 m$^{-2}$) and remained similar in the other three sites (Pan de Azúcar: $2.8 \pm 2.9$ no/ 100 m$^{-2}$, La Campana: $5.6 \pm 8.7$ no/ 100 m$^{-2}$, Nahuelbuta: $2.2 \pm 2.1$ no/ 100 m$^{-2}$, S1 Fig in S1 Appendix).

Overall, the pattern of excavated soil volume from arid to humid was hump-shaped for vertebrates (largest in La Campana), whereas for invertebrates we could not determine a clear geographic pattern along the gradient (Fig 2B). In each site, the soil volume excavated by vertebrates was larger. This difference between the two groups of bioturbators was especially clear in the Mediterranean site La Campana (raw mean and standard deviation for invertebrates: $0.00019 \pm 0.00016$ m$^3$ ha$^{-1}$, for vertebrates: $0.06 \pm 0.18$ m$^3$ ha$^{-1}$) and the humid site Nahuelbuta

**Table 2. Ordinary least squares (OLS) and reduced major axis (RMA) regression analyses of the relationships between the depth and mean diameter of the holes and between the excavated soil volume and hole density for invertebrates and vertebrates (all variables $\log_{10}$-transformed).**

| Relation | method | invertebrate | | | | vertebrate | | | |
|---|---|---|---|---|---|---|---|---|---|
|  |  | r | slope | SE | p | r | slope | SE | p |
| Depth and diameter | OLS | 0.31 | 0.729 | 0.053 | <0.001*** | 0.44 | 0.996 | 0.045 | 0.93 |
|  | (mixed model) |  |  |  |  |  |  |  |  |
|  | OLS | 0.32 | 0.628 | 0.057 | <0.001*** | 0.66 | 1.04 | 0.043 | 0.3 |
|  | RMA | 0.61 | 1.97 | 0.057 | <0.001*** | 0.53 | 1.57 | 0.43 | <0.001*** |
| Excavated soil volume and hole density | OLS | 0.55 | 0.933 | 0.10 | 0.5 | 0.66 | 1.76 | 0.13 | <0.001*** |
|  | (mixed model) |  |  |  |  |  |  |  |  |
|  | OLS | 0.69 | 0.955 | 0.10 | 0.64 | 0.81 | 1.77 | 0.13 | <0.001*** |
|  | RMA | 0.48 | 1.38 | 0.10 | <0.001*** | 0.66 | 2.17 | 0.13 | <0.001*** |

A slope of one represents an isometric relationship. Depicted are statistical method, correlation coefficient, slope, standard error (SE) and p-value (p) of the offset. Significant effects are labelled with asterisks: *:<0.1, **:<0.01

***:<0.001. Data from both field campaigns and all sites were used. Further information on the statistical analysis is provided in the Methods section.

(invertebrates: $0.00015 \pm 0.00022$ m³ ha⁻¹, vertebrates: $0.012 \pm 0.02$ m³ ha⁻¹, S1 Fig in S1 Appendix). Correcting the amount of excavated soil volume for the number of holes, the geographic pattern revealed by the residuals was similar to that obtained based on the analysis of the raw data (Fig 2C); thus, the excavated soil volume was larger for vertebrates than for invertebrates, especially large at the two southern sites.

All predictors for the response variable hole density were significant in the GLMM, with the fixed predictors explaining 48% and the random predictor plot number explaining 39% of the variation (AIC = 2030.7, p < 0.001, S3 and S5 Tables in S3 Appendix). The overall hole density was higher in Santa Gracia and Nahuelbuta during the field campaign from March to April than during the field campaign from September to November while in Pan de Azúcar there was no difference between the two seasons (Fig 3A). For invertebrates, hole density decreased at all sites with increasing vegetation cover. The hole density of vertebrates was

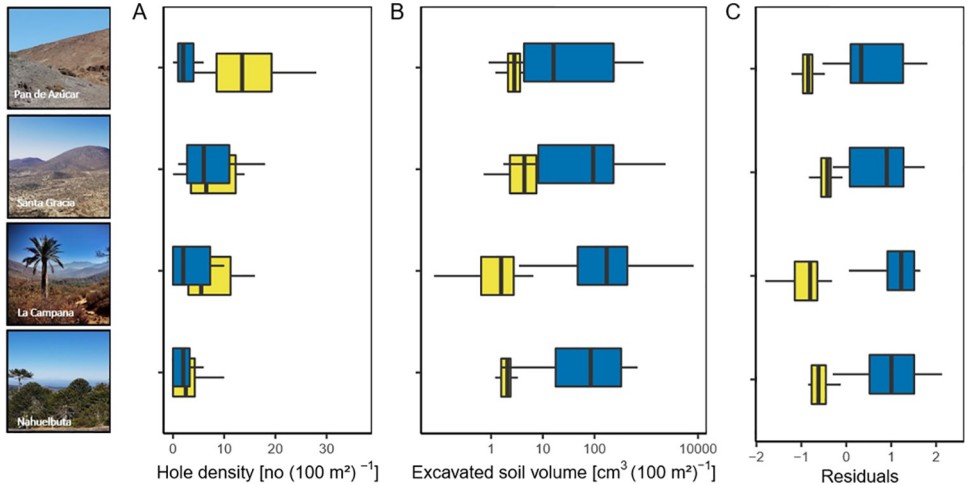

**Fig 2. Bioturbation patterns of invertebrates (yellow) and vertebrates (blue) in each site (Pan de Azúcar, Santa Gracia, La Campana, Nahuelbuta).** (A) Median hole density based on the raw data, (B) median excavated soil volume of holes, (C) the residuals of the excavated soil volume ($\log_{10}$-transformed) after correcting for hole density ($\log_{10}$-transformed) using separate regressions for the two animal groups. Note that the x-axis in (B) and (C) was $\log_{10}$-scaled. Data from the field campaign from September to November were used.

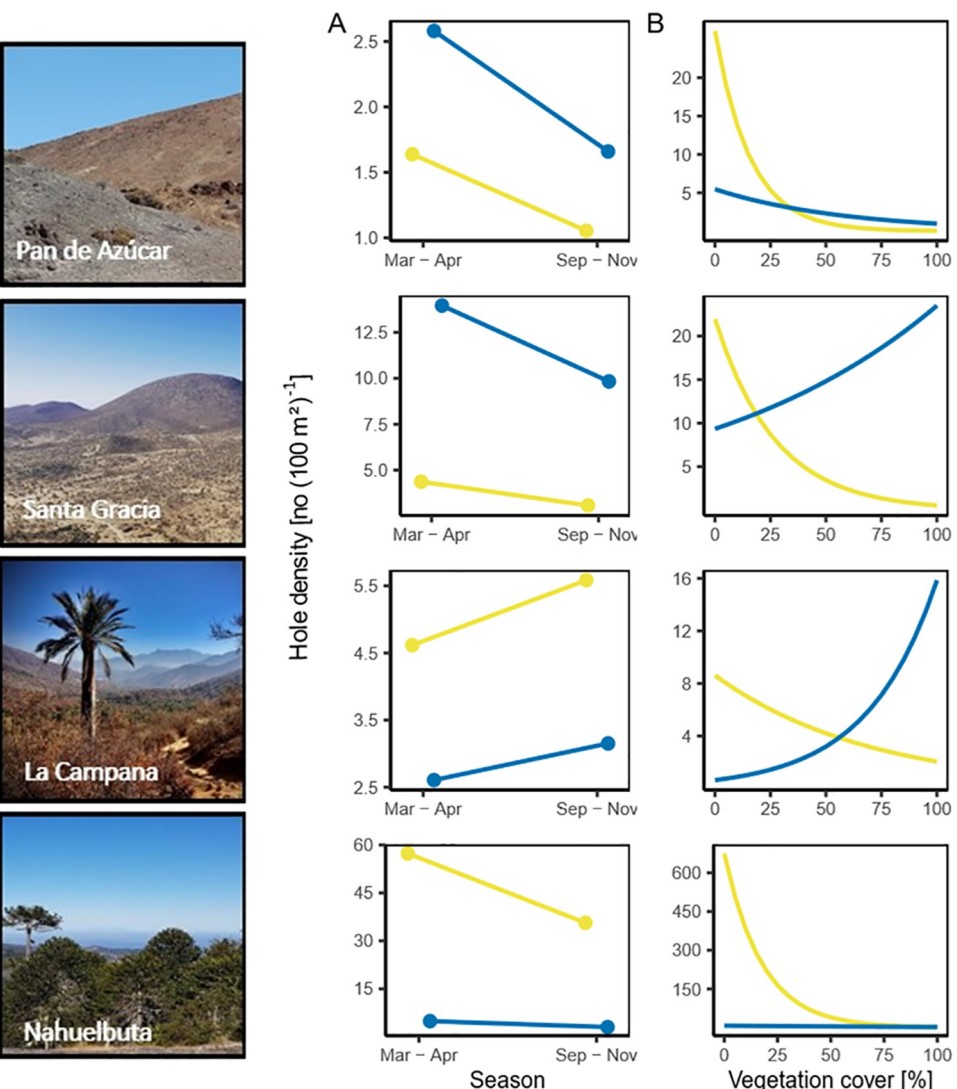

**Fig 3. Fitted relationship between the hole density and fixed effects for invertebrates (yellow) and vertebrates (blue) at each site (Pan de Azúcar, Santa Gracia, La Campana, Nahuelbuta).** (A) Season (autumn: March-April/spring: September-November), (B) vegetation cover [%]. Data from both field campaigns were used.

positively associated with increasing vegetation cover in Santa Gracia and La Campana (Fig 3B). Overall, there was no clear trend in the relationship between the hole density of invertebrates and increasing vegetation cover whereas vertebrates' hole density increased with increasing vegetation cover (S2A Fig in S3 Appendix).

After the exclusion of non-significant independent variables, the fixed predictors season, vegetation cover, hole density and hillside elevation within the fitted GLMM for excavated soil volume explained 85% of the model variation (AIC = 296.67, p < 0.001, S4 and S6 Tables in S3 Appendix). The patterns of excavated soil volume varied for invertebrates and vertebrates with increasing vegetation cover along the climate gradient (Fig 4A). The raw data revealed another trend, as the excavated soil volume increased with increasing vegetation cover for both, invertebrates and vertebrates (S2B Fig in S3 Appendix). In addition, the excavated soil volume increased disproportionally with increasing hole density, with a larger increase for vertebrates than for invertebrates (Fig 4B).

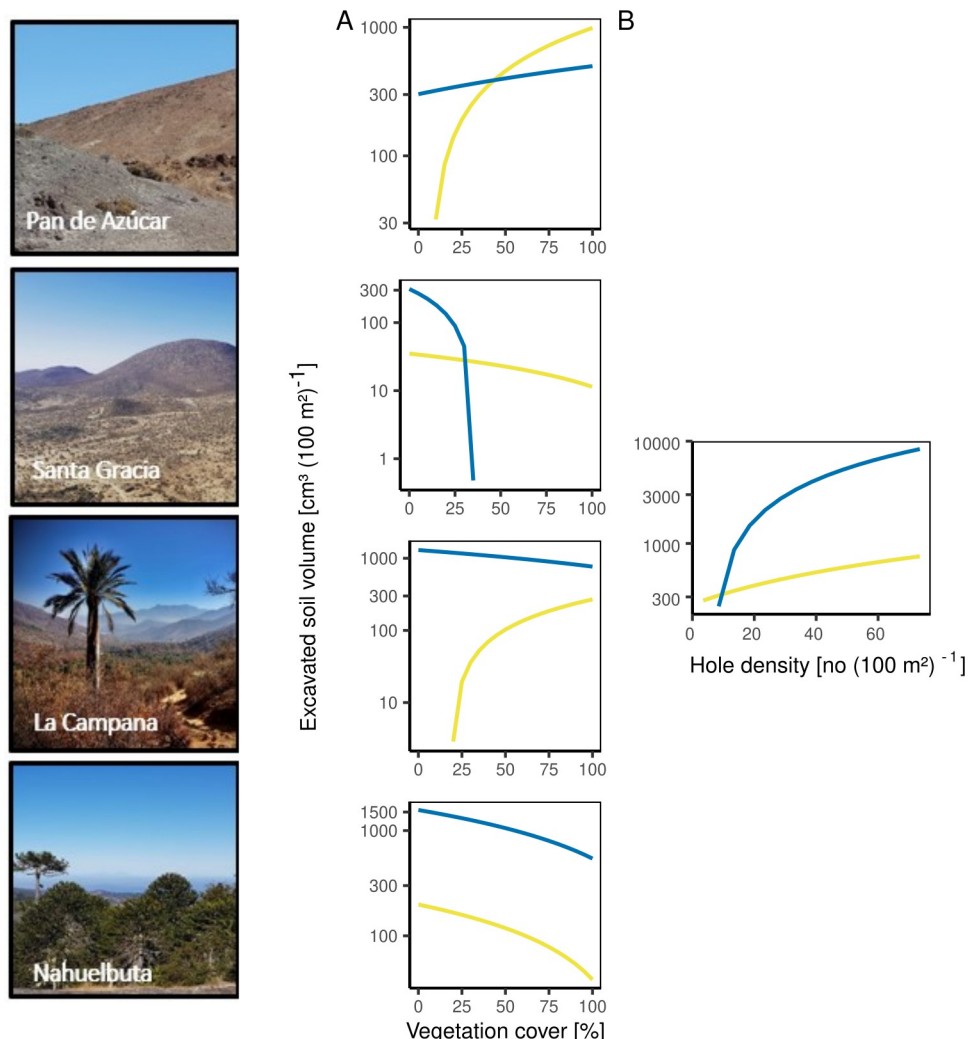

**Fig 4. Fitted relation between excavated soil volume (log$_{10}$-transformed) and fixed effects for invertebrates (yellow) and vertebrates (blue) at each site (Pan de Azúcar, Santa Gracia, La Campana, Nahuelbuta).** (A) Vegetation cover [%], (B) hole density (log$_{10}$-transformed). Data from both field campaigns were used.

## Discussion

Our study showed that while hole density decreased from arid to humid environments, no clear pattern could be discerned for the excavated soil volume along the investigated environmental gradient. However, the contribution of vertebrates to excavated soil volume was larger than that of invertebrates. For the latter, the relationship to the vegetation cover along the climate gradient was consistently negative while for vertebrates it was partly positive.

Before discussing the general results, a few comments should be made on the allometric relationships of the burrow characteristics. Based on the RMA analysis regressions, these relationships were not isometric, as the relative depth of a burrow increased with the increasing diameter of the entrance. This finding suggests that, for bioturbators, larger animals dig deeper into the soil [51]. This relationship presumably reflects the anti-predator behavior of larger animals: with increasing body size animals want to keep their entrance as small as possible to exclude predators [52], but on the same time minimize burrowing cost [53] on one hand, but

have a comfortable nest site [51] on the other hand. Our data do not allow a test of this hypothesis, but further analyses of these allometric relationships are likely to provide a rich source of biological and behavioral information, particularly in studies comparing a large number of animal groups.

Our first hypothesis, that bioturbating activity decreases from arid to humid regions [21–26], was supported by our results for invertebrates, as their hole density decreased from arid to humid climates. Vertebrates, however, created fewer holes in arid than in semi-arid regions. Burrowing vertebrates are, on average, larger than invertebrates [51] such that fewer holes are consistent with a decline in animal density with increasing body size [54]. Accordingly, vertebrates were presumably less frequent in the arid region of our study than in the other climatic zones, such that fewer vertebrate than invertebrate burrows were present over a given area. Similarly, the higher hole density of invertebrates all along the climate gradient can be attributed to the generally higher abundance of invertebrates [55]. However, it is also the case that most invertebrates create their own new burrows while some vertebrates use previously existing burrows as an energy-saving strategy [56, 57]. In particular, larger animals, in our case vertebrates, invest more energy in burrowing effort than smaller invertebrates. Previous investigations showed that the energy cost of burrowing is directly proportional to the amount of soil moved by the bioturbator. Consequently, larger vertebrates, which need to move larger soil amounts to create a burrow of adequate size, will burrow fewer holes [15].

The excavated soil volume did not follow a clear pattern across the climatic gradient and it differed between invertebrates and vertebrates in our study. Similar results were obtained in a recent study measuring the excavated soil volume of bioturbators along the same environmental gradient [36]. The authors found that the excavated soil volume was greater in the semi-arid ($0.56 \ m^3 \ ha^{-1} \ yr^{-1}$) and Mediterranean ($0.93 \ m^3 \ ha^{-1} \ yr^{-1}$) than in the arid ($0.34 \ m^3 \ ha^{-1} \ yr^{-1}$) and humid ($0.09 \ m^3 \ ha^{-1} \ yr^{-1}$) climate zones and that the excavation rates were higher for vertebrates ($0.01–56 \ m^3 \ ha^{-1} \ yr^{-1}$) than for invertebrates ($0.01–37 \ m^3 \ ha^{-1} \ yr^{-1}$). These findings are in line with several studies showing that, due to their larger body size, vertebrates excavate considerably larger volume of soil ($1–5 \ m^3 \ ha^{-1} \ yr^{-1}$) than invertebrates ($<1 \ m^3 \ ha^{-1} \ yr^{-1}$) [55, 58–64] as well as our findings. Those studies together with our own demonstrate the importance of vertebrates as bioturbators along a climate gradient.

Our second hypothesis, that bioturbation activity and quantity respond to seasonal changes [23, 29], was supported by the higher hole density during autumn than spring of the southern hemisphere, as observed at both the semi-arid and humid site. In the arid desert, with a consistent lack of rainfall events, there was no difference between seasons. This is in agreement with previous studies and with the observation that in the southern hemisphere the bioturbation season ends in autumn [65]. Moreover, the climate in Central Chile during the study period in 2019 was drier than usual [66], which may have lessened the differences in bioturbation activity and quantity between seasons. While the relationship between seasons and bioturbation patterns is no doubt, our study suggests that, at least in Chile, the impact of bioturbation is largest in semi-arid and humid climate zones after the autumn rainfall.

The absence of a clear trend between vegetation cover and either bioturbation activity or quantity along the climate gradient was consistent with previous studies examining the distribution of burrow entrances as a function of vegetation [67, 68]. However, we were able to show that the bioturbation patterns of invertebrates and vertebrates differed. The consistently negatively association of invertebrates with vegetation cover supported our hypothesis that some invertebrates are entirely independent of surface resources due to their permanently belowground lifestyle [34]. By contrast, because vertebrates rely on a resource supply from the surface [33], a positive association with vegetation cover occurred only in the middle of the geographic gradient, as in the arid region the vegetation cover is sparse. Vertebrates living in

regions of extreme temperatures characterized by limited resource must invest their energy in digging for food as well as shelter from extreme temperatures in such resource-limited habitats [69]. Furthermore, there is often no vegetation near freshly created burrows, because burrowing typically destroys the vegetation at and possibly adjacent to the burrow [39, 70]. This may have introduced a biased estimate of vegetation cover within plots with fresh burrows and would explain the absence of either a positive or a negative association between burrowing vertebrates and vegetation cover in the humid region. Nonetheless, in general, vegetation cover was shown to be positively associated with vertebrates with a complex influence on bioturbation patterns along the climate gradient.

## Conclusion

Our study showed that climatic conditions and vegetation cover drive the activity and quantity of bioturbation as well as the amount of burrowing by different animal groups. The contribution of vertebrates to bioturbation quantity is large and only bioturbating vertebrates had a positive association, albeit a partial one, with vegetation cover. In its examination of the interaction of abiotic and biotic components, our study demonstrated the intricate relationships between climate, vegetation and the contribution of bioturbating invertebrates and vertebrates. These results provide further insights into the patterns that occur along broad climatic gradients and therefore into the impact of ecosystem engineers on ecosystem processes such as sediment transport, soil water cycling and nutrient availability. In a further study, we will therefore compare physical and chemical soil properties in areas with soil affected and unaffected by bioturbation along the same climatic gradient. Additionally, our findings support the importance of examining impacts of bioturbation on ecosystem processes on a broader climatic scale and thereby encourage similar further studies like the assessment of sediment redistribution rates caused by bioturbation [71].

## Supporting information

**S1 Appendix.**
(DOCX)

**S2 Appendix.**
(DOCX)

**S3 Appendix.**
(DOCX)

**S4 Appendix.**
(DOCX)

## Acknowledgments

We thank the Chilean National Forest Corporation (CONAF) for their support during our field campaigns. Rafaella Canessa provided valuable comments on the statistical analyses and manuscript. We express our gratitude to Robin Fischer and Alexander Klug who participated during the field work.

## Author Contributions

**Conceptualization:** Roland Brandl, Jörg Bendix, Annegret Larsen, Nina Farwig.

**Data curation:** Diana Kraus.

**Formal analysis:** Diana Kraus.

**Funding acquisition:** Roland Brandl, Jörg Bendix, Annegret Larsen, Nina Farwig.

**Investigation:** Diana Kraus, Sebastian Achilles, Paulina Grigusova.

**Methodology:** Diana Kraus, Roland Brandl, Jörg Bendix, Annegret Larsen, Kirstin Übernickel, Nina Farwig.

**Project administration:** Roland Brandl, Jörg Bendix, Annegret Larsen, Patricio Pliscoff, Kirstin Übernickel, Nina Farwig.

**Resources:** Roland Brandl, Jörg Bendix, Annegret Larsen, Nina Farwig.

**Supervision:** Roland Brandl, Nina Farwig.

**Visualization:** Diana Kraus.

**Writing – original draft:** Diana Kraus.

**Writing – review & editing:** Diana Kraus, Roland Brandl, Sebastian Achilles, Jörg Bendix, Paulina Grigusova, Annegret Larsen, Patricio Pliscoff, Kirstin Übernickel, Nina Farwig.

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
