## [Decision Letter · Decision Letter 0]

24 Jan 2022

PONE-D-21-39015Vegetation and vertebrate abundance as drivers of bioturbation patterns along a climate gradientPLOS ONE

Dear Dr. Kraus

Thank you for submitting your manuscript to PLOS ONE. After careful consideration, we feel that it has merit but does not fully meet PLOS ONE’s publication criteria as it currently stands. Therefore, we invite you to submit a revised version of the manuscript that addresses the points raised during the review process. Please submit your revised manuscript by 24 Feb 2022 If you will need more time than this to complete your revisions, please reply to this message or contact the journal office at plosone@plos.org. Please include the following items when submitting your revised manuscript:A rebuttal letter that responds to each point raised by the academic editor and reviewer(s). You should upload this letter as a separate file labeled 'Response to Reviewers'.A marked-up copy of your manuscript that highlights changes made to the original version. You should upload this as a separate file labeled 'Revised Manuscript with Track Changes'.An unmarked version of your revised paper without tracked changes. You should upload this as a separate file labeled 'Manuscript'.

We look forward to receiving your revised manuscript.

Kind regards,

Tunira Bhadauria, Ph.D.

Academic Editor

PLOS ONE

Journal Requirements:

"This study was funded by the German Science Foundation DFG Priority Program SPP 1803: EarthShape: Earth Surface Shaping by Biota, sub-project “Effects of bioturbation on rates of vertical and horizontal sediment and nutrient fluxes” [grant numbers BE1780/52-1, LA3521/1-1, FA 925/12-1, BR 1293-18-1]. We thank the Chilean National Forest Corporation (CONAF). Rafaella Canessa provided valuable comments on the statistical analyses and manuscript. We express our gratitude to Robin Fischer and Alexander Klug who participated during the field work."

We note that you have provided funding information. However, funding information should not appear in the Acknowledgments section or other areas of your manuscript. We will only publish funding information present in the Funding Statement section of the online submission form. 

"This study was funded by the German Science Foundation DFG Priority Program SPP 1803: EarthShape: Earth Surface Shaping by Biota, sub-project “Effects of bioturbation on rates of vertical and horizontal sediment and nutrient fluxes” [grant numbers BE1780/52-1, LA3521/1-1, FA 925/12-1, BR 1293-18-1]."

4. We note that Figures 2, 3 and 4 in your submission contain copyrighted images. All PLOS content is published under the Creative Commons Attribution License (CC BY 4.0), which means that the manuscript, images, and Supporting Information files will be freely available online, and any third party is permitted to access, download, copy, distribute, and use these materials in any way, even commercially, with proper attribution. For more information, see our copyright guidelines: http://journals.plos.org/plosone/s/licenses-and-copyright.

a. You may seek permission from the original copyright holder of Figures 2, 3 and 4 to publish the content specifically under the CC BY 4.0 license. 

Additional Editor Comments:

The authors have done an excellent job of examining the effects of meteorological conditions and vegetation cover on bioturbation activity and quantity, as well as the amount of borrowing by various animal groups. The study also highlighted the complex interactions and linkages between abiotic and biotic components, such as climate, vegetation, and the role of bioturbating invertebrates and vertebrates. However, there are several criticisms and suggestions made by reviewer number one that require clarity, and those remarks must be addressed. As a result, before the paper is accepted for publication, I strongly advise a thorough revision.

Reviewers' comments:

Reviewer's Responses to Questions

**Comments to the Author**

1. Is the manuscript technically sound, and do the data support the conclusions?

Reviewer #1: Yes

Reviewer #2: Partly

2. Has the statistical analysis been performed appropriately and rigorously? 

Reviewer #1: Yes

Reviewer #2: N/A

3. Have the authors made all data underlying the findings in their manuscript fully available?

Reviewer #1: Yes

Reviewer #2: Yes

4. Is the manuscript presented in an intelligible fashion and written in standard English?

Reviewer #1: Yes

Reviewer #2: Yes

5. Review Comments to the Author

Reviewer #1: The study is totally unexplored and uncovered. The hypothesis are excellent and executed very well. In short, impressed with the work done. The only lacking thing i found in MS is future perspective of the study. Therefore, authors can look into this.

Reviewer #2: Comments to the authors

The title is required to undergo some changes. The proposed title is “Study of relationships between bioturbators and their environment in Chile”.

Soil texture depends on the proportions of soil components. Sandy loams are easier to excavate. Hence the second hypothesis needs to be revised.

The research gap is not well understood. With a wide range of microhabitats as well as diversity of life forms on earth along with diversified evolutionary histories, modes of adaptation to microclimates, morphology and behaviour, it is assumed naturally that a pattern in bioturbation activities will be hardly noticed. A detailed review is already done by Kirstin et al. (https://bg.copernicus.org/preprints/bg-2021-75/bg-2021-75.pdf). Hence, it seems like a repetition of what is already done.

Line no. 112: Write the number of quadrats in words to avoid confusion. How was the study sites selected? What kinds of burrowing animals were present at the studied locations? How did you standardize the quadrat size and number?

The reasons behind the findings are very much obvious and predictable. But where is the significance or implication of this study?

6. PLOS authors have the option to publish the peer review history of their article (what does this mean?). If published, this will include your full peer review and any attached files.

Reviewer #1: No

Reviewer #2: No

---

## [Author Response · Author response to Decision Letter 0]

7 Feb 2022

Dear Ph.D. Tunira Bhadauria, 

Please find enclosed our detailed responses to the reviewers’ and editor’s comments for the manuscript PONE-D-21-39015, “Vegetation and vertebrate abundance as drivers of bioturbation patterns along a climate gradient”. The reviewers’ comments were very constructive and mainly concerned (1) the description of the research gap, (2) differentiation of previous work in the study area and (3) future perspectives in the field of research. We have dealt with all of the critical comments in full, and thoroughly revised the manuscript. We hope that you agree that we have satisfactorily dealt with all of the reviewers’ comments in full and that the manuscript is now suitable for publication in PLOS ONE. 

Sincerely,

Diana Kraus on behalf of all authors

Response to reviewers

Thanks to all the reviewers and the editor for your constructive suggestions. We addressed your comments and give a detailed list of changes below.

Apologies for not adequately meeting the PLOS ONE guidelines. We carefully checked all style requirements and now renamed our files according to PLOS ONE guidelines.

Many thanks for pointing us to this important issue. We now explicitly mention the permit in the lines 111 to 113.

"This study was funded by the German Science Foundation DFG Priority Program SPP 1803: EarthShape: Earth Surface Shaping by Biota, sub-project “Effects of bioturbation on rates of vertical and horizontal sediment and nutrient fluxes” [grant numbers BE1780/52-1, LA3521/1-1, FA 925/12-1, BR 1293-18-1]. We thank the Chilean National Forest Corporation (CONAF). Rafaella Canessa provided valuable comments on the statistical analyses and manuscript. We express our gratitude to Robin Fischer and Alexander Klug who participated during the field work."

We note that you have provided funding information. However, funding information should not appear in the Acknowledgments section or other areas of your manuscript. We will only publish funding information present in the Funding Statement section of the online submission form. 

"This study was funded by the German Science Foundation DFG Priority Program SPP 1803: EarthShape: Earth Surface Shaping by Biota, sub-project “Effects of bioturbation on rates of vertical and horizontal sediment and nutrient fluxes” [grant numbers BE1780/52-1, LA3521/1-1, FA 925/12-1, BR 1293-18-1]."

Apologies for the double mentioning the funding information in the manuscript. We now mention the funding information only in the correct position. 

4. We note that Figures 2, 3 and 4 in your submission contain copyrighted images. All PLOS content is published under the Creative Commons Attribution License (CC BY 4.0), which means that the manuscript, images, and Supporting Information files will be freely available online, and any third party is permitted to access, download, copy, distribute, and use these materials in any way, even commercially, with proper attribution. For more information, see our copyright guidelines:http://journals.plos.org/plosone/s/licenses-and-copyright.

a. You may seek permission from the original copyright holder of Figures 2, 3 and 4 to publish the content specifically under the CC BY 4.0 license. 

Many thanks for pointing us to the copyright issue. The images used in the figures 2, 3, and 4 are photos that were taken by the first author of the submitted manuscript. Therefore, Diana Kraus provides the Content Permission Form as an “Other” file with the submission. The photo was uploaded to a server of the university to create the link for the Content Permission Form: 

https://hessenbox.uni-marburg.de/getlink/fiSgU3uz9ud9a4qzPcRM7Dtt/Chile_photos.jpg.

Reviewers' comments:

Reviewer's Responses to Questions

Comments to the Author

1. Is the manuscript technically sound, and do the data support the conclusions?

Reviewer #1: Yes

Reviewer #2: Partly

Many thanks for this generally positive feedback on our manuscript.________________________________________

2. Has the statistical analysis been performed appropriately and rigorously? 

Reviewer #1: Yes

Reviewer #2: N/A

We appreciate the general agreement with our statistical approach.

3. Have the authors made all data underlying the findings in their manuscript fully available?

Reviewer #1: Yes

Reviewer #2: Yes

We are in favor of an open data policy and thus naturally make our data publicly available.

4. Is the manuscript presented in an intelligible fashion and written in standard English?

Reviewer #1: Yes

Reviewer #2: Yes

Many thanks for the positive feedback.

5. Review Comments to the Author

Reviewer #1: The study is totally unexplored and uncovered. The hypothesis are excellent and executed very well. In short, impressed with the work done. The only lacking thing i found in MS is future perspective of the study. Therefore, authors can look into this.

We appreciate the reviewer’s assessment and now included future perspective of the research field in the conclusion statement (lines 333 to 339).

Reviewer #2: Comments to the authors

The title is required to undergo some changes. The proposed title is “Study of relationships between bioturbators and their environment in Chile”.

We appreciate your alternative suggestion for the title of our manuscript. However, we have decided to stick to our original title as it already indicates our major finding: “Vegetation and vertebrate abundance as drivers of bioturbation patterns along a climate gradient”.

Soil texture depends on the proportions of soil components. Sandy loams are easier to excavate. Hence the second hypothesis needs to be revised.

The reviewer is correct in stating that sandy loams are easier to excavate than soil dominated by clay. We thus currently analyze the soil texture across the climate gradient to shed more light on the relationship between bioturbation and soil texture. In our current work we are focusing on the influence of rainfall events on bioturbation activity and thus excluded the misleading term “texture”. We apologize for this lack of clarity. 

The research gap is not well understood. With a wide range of microhabitats as well as diversity of life forms on earth along with diversified evolutionary histories, modes of adaptation to microclimates, morphology and behaviour, it is assumed naturally that a pattern in bioturbation activities will be hardly noticed.

We appreciate the notion that bioturbation patterns could be relatively similar along the climate gradient due to adaptation of life forms on earth. However, previous studies, e.g., Wilkinson et al. (2009) highlighted already that patterns of bioturbation activity change along the climate gradient. To take the reviewer’s point into account, we now added further evidence such as a recent paper by Corenblit et al. (2021) showing that burrowing animals are not only adapting to the existing environment but also co-construct their physical environment (lines 55 to 57). After conducting our study and analyzing the data, we saw variation between research sites in different climate regions and presented it in our results even though it was not as severe.

A detailed review is already done by Kirstin et al. (https://bg.copernicus.org/preprints/bg-2021-75/bg-2021-75.pdf). Hence, it seems like a repetition of what is already done.

Of course, we are aware of the review article by Übernickel et al. who exemplary collected data on bioturbation along the climate gradient of the EarthShape consortium which is part of a larger summary of literature research of burrowing vertebrates and invertebrates. These data were collected on a limited number of plots in each research site and thus gave a first insight into bioturbation patterns. We now collected independent data across the climate gradient that comprised 1) a larger spatial scale with an adequate number of plots per research site and 2) a temporal/seasonal component. The larger sample size enabled us 1) to compare our data with the pilot study of Übernickel et al. and thus to generalize bioturbation activity patterns along the climate gradient and 2) to depict these activity patterns across time and thus grain insights into temporal significance of bioturbation.

Line no. 112: Write the number of quadrats in words to avoid confusion. 

We appreciate your comment, yet we decided to follow the guidelines in academic writing stating that numbers up to nine should always be written in words, anything higher than nine can be written in numerals. 

How was the study sites selected?

Many thanks for mentioning this point. The study sites were preselected for the priority program EarthShape of the German Science Foundation (https://esdynamics.geo.uni-tuebingen.de/earthshape/index.php?id=129) which tackles the overarching research question how microorganisms, animals, and plants influence the shape and development of the Earth’s surface over time scales from the present-day to the distant geologic past. In our project we made use of the large climatic gradient by establishing 20 representative plots per study site to assess the activity of bioturbation. We now slightly reworded this part in the methods section to make the general context clearer. It now reads (lines 105 to 108): “Our study was conducted at four sites representing a climate gradient along the coastal range of Chile (26°S-38°S), extending from an arid desert with a mean annual temperature of 16.8 °C and mean annual precipitation of 12 mm to a temperate humid rainforest with a mean annual temperature of 6.6 °C and mean annual precipitation of 1469 mm.” 

What kinds of burrowing animals were present at the studied locations?

We appreciate this important point. We had planned parallel to the assessment of activity patterns also to capture bioturbating animals in the study areas. Unfortunately, the trapping success was rather limited. This is the reason why we were only able to differentiate the bioturbation activities between vertebrates and invertebrates. To still enable the readers to learn which burrowing animals there are in Chile, we now refer to the study of Übernickel et al., who compiled a literature research on this topic (lines 83 to 84).

How did you standardize the quadrat size and number?

We apologize for the lack of clarity in the methods. We now reworded the explanation how we standardized the quadrat size and number to make this clearer (lines 122 and 123): “The 20 plots per research site were evenly distributed across two opposing hillsides, 10 on the north- and 10 on the south-facing hillslope. “ 

The reasons behind the findings are very much obvious and predictable. But where is the significance or implication of this study?

We appreciate your comment and now explain in more detail the significance and implication of our study (lines 329 and 334): “In its examination of the interaction of abiotic and biotic components, our study demonstrated the intricate relationships between climate, vegetation and the contribution of bioturbating invertebrates and vertebrates. These results provide further insights into the patterns that occur along broad climatic gradients and therefore into the impact of ecosystem engineers on ecosystem processes such as sediment transport, soil water cycling and nutrient availability.”________________________________________

---

## [Editor Report · Decision Letter 1]

10 Feb 2022

Vegetation and vertebrate abundance as drivers of bioturbation patterns along a climate gradient

PONE-D-21-39015R1

Dear Dr. Kraus

We’re pleased to inform you that your manuscript has been judged scientifically suitable for publication and will be formally accepted for publication once it meets all outstanding technical requirements.

Kind regards,

Tunira Bhadauria, Ph.D.

Academic Editor

PLOS ONE

Additional Editor Comments (optional):

After reading the authors' amended article and their responses to the reviewers' comments/suggestions, I believe the work has sufficient scientific content to be approved for publication in its current form. As a result, I recommend that the paper be accepted in its current form for publication.
---

## [Editor Report · Acceptance letter]

15 Feb 2022

PONE-D-21-39015R1 

Vegetation and vertebrate abundance as drivers of bioturbation patterns along a climate gradient 

Dear Dr. Kraus:

I'm pleased to inform you that your manuscript has been deemed suitable for publication in PLOS ONE. Congratulations! Your manuscript is now with our production department. 

Kind regards, 

on behalf of

Dr. Tunira Bhadauria 

Academic Editor

PLOS ONE